# A Review of Factors Affecting *Ganoderma* Basal Stem Rot Disease Progress in Oil Palm

**DOI:** 10.3390/plants11192462

**Published:** 2022-09-21

**Authors:** Nur Aliyah Jazuli, Assis Kamu, Khim Phin Chong, Darmesah Gabda, Affendy Hassan, Idris Abu Seman, Chong Mun Ho

**Affiliations:** 1Faculty of Science and Natural Resources, Universiti Malaysia Sabah, Kota Kinabalu 88400, Sabah, Malaysia; 2Malaysian Palm Oil Board, Kajang 43000, Selangor, Malaysia

**Keywords:** *Ganoderma* basal stem rot, oil palm disease, disease progress, plant disease triangle

## Abstract

In recent years, oil palm has grown on a major scale as it is a prominent commodity crop that contributes the most to almost every producing country’s gross domestic product (GDP). Nonetheless, existing threats such as the *Ganoderma* basal stem rot (BSR) disease have been deteriorating the oil palm plantations and suitable actions to overcome the issue are still being investigated. The BSR disease progression in oil palm is being studied using the disease progression through the plant disease triangle idea. This concept looks at all potential elements that could affect the transmission and development of the disease. The elements include pathogenic, with their mode of infection in each studied factor.

## 1. Introduction

Oil palm (*Elaeis guineensis*) is cultivated widely and mainly in Asia, Africa, and Latin America regions with Malaysia and Indonesia as some of the top producers followed by Thailand, Colombia, and Nigeria. The world production of palm oil has hit around 72.27 million metric tons from the year 2020 to 2021, with Malaysia and Indonesia as the leading exporters of palm oil globally [1]. The yearly addition of oil palm plantations, and, thus, palm oil production, can be linked to the increasing demand for foods and industries and the fact that oil palm is most efficient when it comes to comparing commodities’ yields and other benefits. Consequently, these have caused the plantation schemes for oil palm to expand over the years.

Research and validation had been made on the extension of oil palm plantation areas in Indonesia, Malaysia, and Thailand through *Copernicus Sentinel*-1 microwave backscattering time series, Landsat multispectral time series, and auxiliary dataset methods [2]. The findings proved that regions in Sumatra, Kalimantan, Peninsular, and Insular Malaysia along with Thailand had all expanded their oil palm plantation by 40 percent with respect to their land. One of the factors that encouraged the expansion of oil palm plantations in Malaysia was the previously existing areas for planting oil palm and the suitable environmental conditions for growing oil palm where most of its land area has high biophysical suitability for the crop [3].

Oil palm is grown on a massive scale and is mostly suitable for certain regions in Malaysia, making it a well-known commodity crop in the country. However, existing threats may deteriorate the oil palm plantations if further actions are not conducted to overcome the issue. As of now, one of the issues faced by oil palm major stakeholders and smallholders is the *Ganoderma* basal stem rot (BSR) disease. Two major diseases that threatened most of the oil palm industry are known as basal stem rot (BSR) disease and lethal bud rot (fatal yellowing) disease [4,5]. Other oil palm diseases include *Fusarium oxysporum f.* sp. *elaeidis*, upper stem rot (USR), red ring, and sudden wilt [6]. The BSR disease is caused by the white-rot fungi of the *Ganoderma* spp., and to date, almost 60 percent of oil palm plantations in the country are affected by it [7]. BSR disease is harmful as most infected oil palm trees will usually stop producing fruits and eventually collapse.

*Ganoderma* is a parasite that lives saprophytically on a food base obtained from stumps and roots. If the pathogen is not detected early and control measures are not taken, the infected trees may be dead within six to 24 months for immature palms and one to two years for mature palms after the symptoms have developed [8]. Some of the affected trees, however, can live up to a few more years. The disease is believed to be able to cause the death of up to 80% of plantings halfway to their economic life [9]. Ground surveillance of BSR caused by pathogenic fungus *Ganoderma* in oil palm trees was applied to determine the status of incidence and distribution of disease among the participating replanting incentive scheme smallholders [10]. BSR disease was found to affect mostly the oil palm trees of three states in Malaysia: Johor (1032.96 ha with 487 smallholders), Sabah (930.85 ha with 252 smallholders), and Perak (718.49 ha with 410 smallholders).

## 2. Disease Progress

### 2.1. Plant Disease Triangle

A plant disease is defined as any disturbance to the plant’s normal physiology caused by an agent, which results in an altered appearance and/or decreased productivity compared to a healthy, normal plant of the same variety [11]. Pathologists have discovered over time that the interactions of three main components have a crucial role in the development of disease in a plant population depicted by a disease triangle. The disease triangle concept plays an important role when it comes to studying disease progress within plants. The concepts in the disease triangle may be utilized as a basic principle in the search for the root cause of the disease and its complementary factors [12,13].

The three fundamental factors involved in the disease triangle are the existence of disease caused by interaction with the host, a virulent pathogen, and an environment that encourages the development of the particular disease. Thus, controlling any one of the factors is believed to help mitigate the disease in plants. For instance, pathogens can be controlled by the use of pesticides, cultural techniques, or other practices; the environment, however, is natural and cannot be changed by human action; plant vulnerability can be reduced through the use of cultivars of disease-tolerant or -resistant plants [14].

The inclusion of human factors based on the influence of human activities and management practices should not be neglected, as that can affect the occurrence and severity of a particular plant disease. However, the three main aspects already have a certain degree of human influence and, thus, the disease triangle itself is ample as a framework to discuss the diverse factors affecting the particular disease. The concept is aligned with a study made by Scholthof [15], where she affirmed that the disease triangle is a conceptual model that demonstrates interactions between a susceptible host, a virulent pathogen, and a favorable environment. Hence, the model can be used to predict epidemiological events in plant health in both local and global communities.

Diseases in plants are divided into biotic and abiotic [16]. The biotic disease involves disorders caused by living organisms such as viruses, fungi, bacteria, insects, mites, and nematodes, while the abiotic disease is caused by nonliving factors such as drought, over-watering, nutrient deficiencies, improper cultural practices, or chemical injury. A fungus is the main disease-causing agent in almost all plant illnesses. All of the economically significant plant diseases, except those brought on by nematodes, are brought on by fungi [11].

### 2.2. The Assessment of Disease Progress in Plants

In order to create sustainable and efficient disease management techniques, the main goal is therefore to improve the understanding of how diseases spread throughout populations of host crops and how other factors may affect this development. The level of disease existing in a population of plants is often tested numerous times to track temporal disease development. A disease progress curve, which essentially illustrates the dynamics of a disease’s progress through time, can be created using a collection of data. The results of dynamic interactions between the host, pathogen, environment, and crop management can be shown by a simplified disease progress curve. Hence, to comprehend the temporal progression of plant diseases, numerous novel analytical and modeling approaches have been applied to plant disease epidemiology [17].

One case example is a study describing the disease progress curve of the cocoa black pod through a comparison of nonlinear models [18]. The disease progress curve over time is crucial to be recognized so researchers or planters can determine the resistance level of cocoa against the *Phytophthora palmivora* disease. They applied and compared exponential, monomolecular, logistic, and *Gompertz* models using the Akaike Information Criterion (AIC) and lowest Bayesian Information Criterion (BIC) goodness-of-fit tests to select the best-fitted model. The results gained showed that the rate of disease severity could be fitted to a *Gompertz* curve because it showed the lowest values of AIC and BIC. The particular finding is convenient for predicting the disease severity and determination of resistance level through estimation of the area under the disease progress curve (AUDPC).

A previous study on *Ganoderma* involved modeling the yield loss of oil palm due to the particular disease by using the backward elimination-based regression method [19]. The best model was developed through residual analysis, and the results revealed that the mean value and standard deviation of the standardized residuals were, respectively, zero and one. The standardized residual distribution was also homoscedastic, normal, and devoid of outliers. Reasonable forecasting was determined by the mean absolute percentage error (MAPE) value, which was used to assess the performance of the best model’s predictions. A limited number of studies have been considered to model the disease progress of *Ganoderma* disease in oil palm, although studies on disease progress through mathematical models have long been applied. In fact, to date, no studies have modeled the disease progress of *Ganoderma* BSR disease based on factors associated with the disease triangle concept.

The assessment of disease has not changed significantly for decades, making it an ideal target for Agriculture 4.0 technology, where simulation modeling and big-data sets are essential components of the current digital era. Paterson [20] simulated the evolution of BSR disease progress over the next 80 years using the Agricultural 4.0 approach and found that there have been no significant innovations in the treatment of diseases, while some remediation techniques such as developing oil palm in novel regions with suitable climate may help lower BSR disease progress. Between the years 2050 and 2070, the climate for oil palm will significantly deteriorate in terms of cold, heat, or dry stress or draught events [21,22], and a rise in disease is projected as a result of the adverse impact of climate on the growth and resistance of oil palm. Therefore, there is a roughly a 30-year duration of opportunities to take corrective measures to address future BSR levels.

The use of nonlinear growth functions and integrated measures such as AUDPC will play an important part in informing suitable tactical and strategic decisions to be made for control treatments [23]. These approaches have proved their usefulness in expressing control effectiveness and optimizing, or changing, the control practices. The liaison between epidemiology, disease management, and a wide range of mathematical modeling has yet to produce a major impact on practical disease management in the future. Thus, this study combines every possible aspect, which is the utilization of plant disease triangle concepts to learn the possible factors that may affect the *Ganoderma* BSR disease progress in oil palm over a certain period. The determinants are divided into host, pathogen, and environmental factors with the inclusion of human and management practices elements.

## 3. Factors Affecting the *Ganoderma* BSR Disease Progress in Oil Palm

### 3.1. Pathogenic Factors

Key features that need to be highlighted when discussing pathogenic factors for plants according to Keane and Kerr [13] are the presence of a pathogen, its level of virulence and aggressiveness, its adaptability, the efficiency of dispersal and survival, and its reproductive strength. The ability of a pathogen to invade and multiply within the host is known as virulence. Disease transpires when a virulent disease pathogen meets a vulnerable host under favorable environmental conditions prone to the development of disease [24,25]. The evolution of pathogens is commonly due to many forces including spatial dispersion, recombination, genetic drift, and selection by host plant resistance [13].

Consequently, for oil palm, major plantations are mostly affected by the BSR disease caused by a soil-borne fungus known as *Ganoderma boninense*, including Malaysia and its neighboring countries. This disease has prevailed through consecutive replanting schemes, infecting younger plantings and diminishing the economic life span of each cropping cycle [26]. The presence of the *Ganoderma* spp. pathogen can be seen from the existence of an active fruiting body, as shown in Figure 1 [27], rotted trunk, unopened spear, hanged skirting, dried front, yellowed front, foliar symptoms, whether there is the presence of bunch or fruits, the disease severity, and the disease incidence.

*Ganoderma boninense* is indicated through basidiocarps that are large, chronic, woody brackets that are lignicolous, leathery, and occasionally with a stem. The form of fruit bodies usually grows in a fan-or-hoof-like form on the tree trunks. They also have double-walled, truncated spores with yellowish to brownish ornamented inner layers. One of the justifications for why the disease has not been well detected beforehand is due to the number of alternative and consecutive events in its cycle. At first, there must be an exposure of the wood through injuries for the cells around the injured zone to be oxidized and discolored because of biochemical changes. This event can encourage microorganisms to land and grow on the opened wound. Next is for bacteria or fungus to live on the wound, thus adding more to the discoloration, wetness of the area, and erosion of the cell wall. Lastly, the wood-rotting fungi will integrate and then digestion of cell wall components will start [6].

Symptoms of oil palm trees affected by *Ganoderma* BSR are water stress, one-sided mottling of the canopy, crown flattening, multiple unopened spears, and production of basidiocarps on the lower stem [5]. Figure 2 [7] shows some of the symptoms mentioned above.

In previous investigations, it has been discovered that many species, including *Ganoderma boninense*, *Ganoderma zonatum*, and *Ganoderma miniatotinctum*, are responsible for the BSR disease of oil palm in Malaysia. In contrast, *Ganoderma tornatum* is nonpathogenic and only infects dead trunks of oil palm trees. The most aggressive known pathogen is *Ganoderma boninense*, according to experts. However, studies have revealed that the dominant *Ganoderma* spp. causing BSR disease in oil palms might vary depending on location. The identification of the pathogen is essential in deciding the best course of treatment for the condition because different species of *Ganoderma* display a range of traits and levels of aggression [28]. High levels of genetic variation have been found in *Ganoderma boninense* monokaryons, suggesting that this species is genetically heterogeneous and that this may be due to outcrossing between isolates over generations [29,30] or to different geographic locations, with the pathogen potentially coming from either the same species or a similar species [31].

The white-rot fungus *Ganoderma* is a member of the class *Agaricomycetes* and the family *Ganodermataceae*. *Ganoderma boninense* is thought to primarily produce inoculum from basidiospores as they are easily spread by wind or animal vectors [32]. The basidiospores develop into monokaryotic vegetative mycelia once they are in a favorable environment. A dikaryotic mycelium is created as a result of nuclear exchange and migration, which invades and establishes itself inside the plant host. Under the right climatic conditions, the dikaryotic mycelia later give rise to the production of the fruiting body. The multicellular reproductive structure known as the fruiting body, or basidiocarp, is where karyogamy takes place and meiotic spores are created. Once basidiospores are produced by the basidiocarp, the sexual cycle of *Ganoderma* is complete. The tetrapolar mating system, which controls sexual reproduction, encourages outbreeding and genetic variation in the same plantation region, creating dynamic populations and thus becoming the main factor contributing to ineffective disease management [32,33].

The most contagious tissues in oil palms were the stumps. After BSR stumps ceased to be contagious, stumps derived from healthy palms became BSR sources and remained infection foci for years. In bait seedlings, symptoms started to show up after six months, with a mean symptom onset period of 14 months. Infection foci developed more gradually in larger stumps, which are 50 cm high compared to smaller stumps, which are 2 cm in height [34]. Usually, pathogens with a mixed reproductive system, a high potential for genotype flow, large effective population numbers, and high mutation rates are most likely to disrupt resistance genes. As for pathogens with strict asexual reproduction, minimal opportunities for gene flow, small effective population numbers, and low mutation rates, they are the ones with the lowest risk [35].

It is crucial to have insights into their mode of infection. Sexual reproduction was believed to hold a significant role in the epidemiology of BSR disease [25]. Insects, wind, and rain are also elements that may carry spores to wounds on trees that usually have been cut. The *Oryctes* beetle [8] and larvae of the *Sufetula* spp. caterpillar hold a small part in spreading the *Ganoderma* spores [36]. Some of the infection modes are in-contact roots with neighboring infected palms and airborne basidiospores. Experiments of releasing a large amount of *Ganoderma* spores in a field had been conducted and researchers found out that not all trees were infected [37]. Thus, it was pointed out that infected tissues within the soil are the reason for the disease to be spread to healthy roots extensively compared to airborne spores. Rees et al. [5] in their study confirmed that the probability of the main mode of infection of the *Ganoderma* BSR disease may be caused through root invasion.

Furthermore, the disease is linked to the decay of the lower stem of the palm, which leads to severe symptoms such as unopened and the flattening of spear leaves [38]. There are three main modes of infection of white-rot fungus *Ganoderma boninense*, which are the inoculum left by alternative host plants, the inoculum from the infected trees spreading mycelial root contact, and also the airborne basidiospores [6]. Basidiospores hold an important role in the physical and genetic discontinuity of BSR infections, especially in the high genetic diversity of *Ganoderma boninense* segregated within plantations, even between most nearby trees with BSR disease [39].

Another study was conducted on spatially identifying the *Ganoderma* disease pattern based on the disease incidence elements, which were the nearest-neighbor analysis, refined nearest-neighbor analysis, and second-order spatial analysis using Ripley’s K function [40]. The findings proved that all the disease incidence elements showed that the distribution of infected palms in the studied areas was clustered. In other words, the study verified that the infected palms’ spatial distribution does not occur randomly, but more in a clustered pattern whereby the spread of the disease developed from tree to tree conceivably through root contacts.

Pokhrel [25] in his findings agreed that population density has a significant impact on the development of a disease. In this review, disease incidence would be used to reflect the population density of infected trees based on the number of trees that develops BSR disease in a particular period. It is important to know which plot of oil palm tree has been affected by the BSR disease and the number of neighboring palms around the particular palm that is infected with the disease. Interpolated density, spatial autocorrelation, and hotspot analysis were conducted on field data to provide descriptions of how data are correlated with distance (a measure of the degree of spatial dependence between samples) and to detect the areas that experienced a high density of BSR diseases using the Kernel density estimation technique. Information on the spatial and temporal pattern of the BSR disease in oil palm plantations is significant to know more about the disease dynamics and develop accurate sampling plants to better assess crop loss concerning the disease intensity. They found out that the incidences of BSR disease are random and higher in a higher-density oil palm plantation compared to those in a lower-density plantation. In addition, the occurrence of the BSR disease is attributed not to an infection from tree to tree but more to the pressure of the disease in the area. Most of the infection spreads continuously in a random manner through root contact [41,42].

### 3.2. Host Factors

The disease can develop within the host when it is susceptible to the particular pathogen that comes in contact with it at its appropriate growth stage. The occurrence of pathogen–host interactions in plants is very definitive as pathogens will only attack specific hosts that enable them to obtain their food and living sources for growth and development [13,25]. The process of infection may either be successful or unsuccessful reliant on the host type, and whether they are susceptible or resisting [43]. If the host is resistant to a pathogen, a disease will not develop even under suitable environmental and setting conditions. The host’s response to pathogens commonly depends on the developmental stage of the host when challenged by the pathogen [44]. In this case, the oil palm acts as the host as it becomes the home or source of living for the *Ganoderma* BSR fungus to live.

Host resistance is often seen as an essential and effective element in preventing and controlling plant diseases because it is comparatively cheap, biologically safe, and practical for farmers [45]. Genetically resistant materials are currently being used to manage BSR disease in oil palm [46,47]. The frequency of natural infections in the field, which takes a long time, has generally been the basis for observations on BSR resistance. The root inoculation procedure can offer a different screening strategy. Ariffin et al. [48] used this method to identify substantial variations in susceptibility among several commercial *Dura* x *Psifera* (DxP) materials.

For example, [49] found that the AVROS oil palm variety, which is the most widely planted, is said to be more resistant to *Ganoderma boninense* than other commercial types, such as Calabar and Ekona, which share the same *Dura* but are *Psifera* of African provenance. Based on the greater levels of ergosterol, which is a particular marker linked to a fungus, discovered in the roots of some Ekona and Calabar varieties and the higher disease severity score compared to AVROS, it has been shown that these varieties are more susceptible to *Ganoderma boninense*. The root inoculation method revealed variations in sensitivity among seedlings of several progenies, but none demonstrated complete resistance. Instead of invasion tolerance, as had been inferred from field observations, this generally linked to changes in the rate of *Ganoderma* spread in tissues of the roots and stem bulb as they were invaded by *Ganoderma boninense* [45,50].

The possibility of the co-evolution of *Ganoderma boninense* within oil palm and coconut crops has suggested that the fungus may be palm-specific-focused and that the level of disease may increase in future planting schemes [51]. The factors of the host that can be examined are the generation of the tree, variety of crops, previous crop planted on the plot, age of the tree (based on its planting year), and source of seedlings. The oil palm plantation in Miri’s selected plot aged 11 years and the first-generation plantation have a higher BSR occurrence compared to the 18-years-old oil palm in Betong’s selected plots [52]. This issue may be due to the history of the estate where the *Ganoderma* species may have existed in the area before the area was cleared for the plantation of oil palm.

A previous survey carried out by Turner [53] on Malaysian oil palm estates resulted in the conclusion that old oil palm trees with ages above 25 years old and younger plantings on sites that had coconut palms as a previous crop were the cause of *Ganoderma* disease infections. Coconut tissues inhabited by *Ganoderma* are the infection center for the disease spread, mostly through root contact with stumps left in the ground. Severe BSR disease was in scattered and widespread form as observed in a studied palm field at Pamol Sabah Malaysia when due for replanting at 25 years [54]. The spread of the disease was rapid at the plot where the old stand was piled and not removed.

The population structure and demographical history of *Ganoderma boninense* that causes BSR disease in oil palm across the oldest known regions of interaction between oil palm and the disease were studied in Peninsular Malaysia and Sumatra [55]. It was found that the spread of the disease was neither caused by planting generation, the genetic background of planting material, nor any sort of physical obstacles. The population of the *Ganoderma boninense* was increasingly linked to ice age periods and not due to the first few years of the industrial oil palm cultivation expansion period.

The intensity of a *Ganoderma* infection was positively connected with palm age. This is because it has to do with the growth of the *Ganoderma* colonization process on accumulative fields, particularly on peatlands that are rich in organic material. Additionally, *Ganoderma* infection progresses gradually, explaining why the disease’s symptoms will not develop until it is advanced and takes a longer period. When the oil palm is old, the majority of the disease’s symptoms become visible as the accumulation of *Ganoderma* inoculum in the field is mostly to blame for the occurrence of a higher prevalence of *Ganoderma* assault on the subsequent generation of oil palm [56,57,58]. The loss brought on by *Ganoderma* had reportedly increased rapidly as a result of the subsequent substantial debris of oil palm tissues that were infected, which added to a significant buildup of field inoculum from the previous crop [59]. Replanting active oil palm trees on smallholders’ lands and estates with a history of the disease increases the danger of the disease’s outbreak.

The rate at which *Ganoderma* disease spreads from estate to estate might differ, suggesting that isolates of *Ganoderma boninense* from various locales have varying levels of aggression. The most important variables that can influence the severity of the disease in treated seedlings are the age of the oil palm seedling and variations in the level of aggressiveness among different *Ganoderma boninense* isolates [60,61]. For instance, Goh et al. [62] studied the level of disease severity in 5-month-old seedlings that had been individually inoculated with twelve different *Ganoderma boninense* isolates. Their findings led to the findings that out of 12 different *Ganoderma boninense* isolates, the isolates or seedlings, BtLintang (Kedah) with 63.3 percent, Fraser (Johor) with 8.3 percent, and Pinji (Perak) with 3.7 percent were the least aggressive, based on their disease severity index value.

BSR disease can kill 80 percent of standing oil palm in an infected area within 15 years of the crop’s establishment [1]. The spores of the *Ganoderma* can spread through wind and water. Furthermore, it can be transmitted via soil or roots of transplanted seedlings from a nursery of any particular location where the plantation was developed. Any land that is affected with BSR disease is recommended to be left empty or used for other types of crops for several years to prevent its spread and severity of future infections. Unfortunately, when the spores or fungus have settled in the soil, replanted palms would eventually capitulate to the disease in their early life cycle.

### 3.3. Environmental Factors and Other Relevant Factors (Human and Management Practices)

Environmental factors are considered as a top impact on plant pathogen development. Serious disease cases will not occur except when the environmental conditions encourage it and are in favor of its development. The environment may affect plant pathogens in terms of their survival, strength, multiplication rate, sporulation, direction, dispersal of inoculum distance, spore germination rate, and penetration [25,63]. The environment can also directly influence the disease incidence and its progress due to the existing interaction effects between the host and pathogens. Significant elements are the temperature, duration and intensity of rainfall and dew, soil temperature, soil water content, soil fertility, soil organic matter content, wind, history of fire for native forests, air pollution, and herbicide damage [13].

Accordingly, environmental factors in relation to the existence of *Ganoderma* BSR disease can be related to the soil type, the topography of the land, the amount of rainfall in the areas (climate change), and the frequency of flooding in the area of oil palm plantations. The ecological environment has a significant impact on the spread of disease [64]. Failure to detect the BSR disease development in the industry is the cause of complications for the next generation of planting regardless of soil type or any environmental factors that might contribute to the occurrence of the disease. Similar findings were made by Rolph et al. [65], whereby the complication in regulating the disease was due to the lack of sufficient data on *Ganoderma* spp. needed for expanding a reliable initial stage diagnosis system. Therefore, various plantation management schemes or practices are being exercised to curtail wounds on trees, including by improving treatment and harvesting activities, and clearance of old trees before extreme age susceptibility [66].

Areas with lateritic soil types were found to be most affected by the *Ganoderma*, followed by coastal, inland, and peat [10]. The BSR illness was infectious in oil palm plantations with all types of soil. Previously, it was believed that all soil types were resistant to *Ganoderma* disease, but this is no longer the case. The coastal regions of Malaysia have been reported to have a greater frequency of BSR disease [53,67]. Likewise, the majority of the soils tested in the coastal regions of Peninsular Malaysia’s western half are susceptible to BSR disease as reports revealed a high prevalence of BSR on oil palms in inland regions, including *Holyrood*, *Sungai Buloh*, *Rasau*, and *Bungor* series [68], *Batu Anam* or *Burian* series, *Munchong* series, peat soils [69], and lateritic soils, particularly Malacca series [70].

Peat soil was once assumed to be noncontributive to the development of BSR disease [8], but it contradicts the findings made by several authors [41,52,71,72,73] where they found out that there exists a BSR disease incidence on oil palm that was planted on peat. Other than that, increasing unfavorable weather will result in concurrent rises in BSR disease [74]. The *Ganoderma* fungi are adaptable and will adjust to future climate change more rapidly than oil palm by adopting strains that are more aggressive toward oil palm, while oil palm will take longer to adapt [55,75]. Some methods to overcome the effect of climate change on oil palms, which, in turn, can control the BSR disease and assist the sustainability of the crop, are to grow the palms in suitable regions outside of Southeast Asia and in areas with low levels of the existing disease, perform cross-breeding on the oil palm, use arbuscular mycorrhizal fungi with and without reduced tillage, or empty fruit bunch application [76].

At least ten distinctive methods of managing the disease have been conducted with varying levels of success, namely, soil mounding, surgery, sanitation or removal of diseased materials, ploughing and harrowing, fallowing, planting of legume cover crops (LCCs), chemical treatments, fertilizers, biological control, and resistant planting materials [6]. Even though managing a field free of pathogens is unattainable, a comprehensive management strategy to minimize the entrance of the pathogen to healthy palms can significantly reduce the disease. This is accomplished by reducing tree wounds, enhancing treatment and harvesting procedures, and removing old trees before they become extremely susceptible to disease.

These factors are counted as human factors as some studies have found that not all of the methods can be utilized to prevent the development of *Ganoderma* BSR disease among oil palm trees. For instance, soil mounding is the method in which the soil is heaped around the trunk to a height of approximately 75 cm to extend the life of the tree, but it was proven to be not effective in regulating the BSR disease [77]. Similar to the planting of ground covers that may invite legume species that are susceptible to *Ganoderma boninense*, injection of fungicides such as hexaconazole is seldom practiced because they have not shown effective control measures on the disease [78]. Surgery on trees by removing dead tissues or basidiocarps either by a chisel or mechanical back-hoe produced average results except for a few successful small plantations [79,80].

Digging trenches around the infected palms had been recommended as a controlling strategy to stop mycelium from proliferating by root contact with nearby healthy palm trees. The expense of digging and maintaining trenches is also costly as it relies on the type of soil being used. Considering mycelial development through root contact is thought to be the cause of BSR infection, and cleanliness during replanting is regarded as a key strategy for reducing BSR. However, the results demonstrated that this strategy only helps in minimizing the illness incidence but did not sufficiently reduce BSR [81]. Other cases that used a chemical treatment to control BSR disease showed that when the oil palm trunk was injected with a mixture of the fungicides carboxin and quintozene, the chemical treatment results indicated a considerable reduction in BSR incidence [82].

Chemical pesticides, however, are bad for the ecosystem because they prevent the growth of beneficial bacteria. Growing environmental concerns and the costly expense of chemicals have, therefore, pushed researchers and farmers to discover nonchemical methods of controlling BSR, such as the use of biological control agents and pathogen-resistant cultivars. Using bio-fungicides to control plant diseases is a desirable agricultural practice as they are biocompatible and biodegradable and it can be carried out without harming the plant or leaving behind poisonous materials that are bad for soil organisms and the environment [83,84,85].

Modern approaches to disease and pest management strongly warn against using chemicals in crops. Biological control was first applied around the end of the 19th century, but similar methods had been in use for at least 2000 years prior [86]. There are four types of biological control, namely, classical, natural, conservation, and augmentative controls. The approach is frequently regarded as an appealing, environmentally friendly substitute for pest control [87,88]. Developing sustainable agriculture at a lower environmental cost has triggered a significant technological, economic, and political discussion about the idea of implementing biocontrol [89]. A preventive strategy that can cut down on pesticide use by about 50% has been set-up in various countries [90]. These actions display a significant understanding of the excessive accumulation of harmful residues in the environment and the numerous connections in the food chain [91]. They also point to the absence of alternatives to lessen the agricultural sector’s dependency on pesticides. In this circumstance, it seems essential to develop a better understanding of biocontrol in order to increase their effectiveness [89].

Malaysia’s oil palm industry, in compliance with certificating the country’s oil palm plantation with the Malaysian Sustainable Palm Oil (MSPO) and Roundtable Sustainable Palm Oil (RSPO), adopts good agricultural practices (GAP), which includes reduced chemical pesticides usage by enhancing the use of biocontrol such as biopesticides application [92]. For instance, the use of chitosan in controlling *Ganoderma* infection in oil palm [93] found that lower concentrations of chitosan had successfully diminished the severity of BSR disease in oil palm seedlings. Researchers in [94] had similarly applied chitosan to formulate chitosan-based agro-nano-fungicides by encapsulating fungicides of hexaconazole and dazomet into the chitosan nanoparticles. In short, it is important to understand how biocontrol agents (BCAs) are effective against specific plant diseases by looking in detail at how the plant, the pathogen, the biocontrol agent, and the environment interact with the biocontrol activity [91]. With this information, risk factors that encourage the selection of plant pathogen strains resistant to BCAs will be identified. Additionally, it will enable the identification of BCAs with a lower risk of failure. It is crucial to select agents that are effective in a range of circumstances, including soil texture, moisture, temperature extremes, and competition, to ensure effective biocontrol [95].

A significant study conducted for 7 years on oil palm-replanting schemes through windrowing, fallowing, and poisoning practices had contributed to the bits of knowledge of *Ganoderma* disease incidence effective measures [96]. The research concluded that there were no significant effects of windrowing system and poisoning palms on the incidence of *Ganoderma*. Nevertheless, fallowing for up to one year had a significant effect on reducing *Ganoderma* incidence in the next generation. Some of the causes of the reduction in infection levels are most likely reduced soil inoculums at the time of replanting and also delayed planting of palms. *Ganoderma* is known to be a poor competitor in nonsterile soil or organic debris and, thus, the inocula of *Ganoderma* that remained in the field, but without the presence of its host and in the presence of antagonistic soil microorganisms, would be decreased.

It is important to address the right measures in controlling the *Ganoderma* BSR disease and anything related to the sustainability of oil palm because the global issue due to expansion of oil palm cultivation is a constantly growing concern. Oil palm planters must improve cultivation methods to support environmental sustainability, such as barring the clearing of rainforest, restoring degraded and infertile lands for oil palm plantations, abstaining from slash-and-burn techniques, protecting the environment and promoting tropical biodiversity by growing wildlife corridors close to or between plantations, and operating plantations morally and legally [97]. Additional themes related to environmental sustainability of oil palm include biodiversity, deforestation, environmental pollution, and peatland conversion [98]. Issues that should not be neglected include site preparation for oil palm plantations resulting in soil erosion, which temporarily increases the amount of silt in receiving streams [99]; peat soils that were drained and burned to create plants, which worsens global warming by releasing the carbon that has been stored there [100]; water pollution caused by leaching of fertilisers and pesticides into waterways by surface runoff, which may affect aquatic ecosystems when oil palm farms are in operation [101].

## 4. Conclusions

BSR disease progress in Malaysia and other palm oil-producing countries shows an alarming increase in both large estates and smallholder plantation areas. This paper reviews all the possible factors affecting BSR disease progress in oil palm caused by the *Ganoderma boninense* pathogen by using the plant disease triangle concept. The disease progress will continue showing increments if no proper control measures and mitigation are conducted. Finding a viable, effective disease control system for BSR disease appears to be difficult and requires a variety of methodologies and strategies given that early detection is still debatable. Furthermore, the issue is vital to avoid further direct and indirect economic losses caused by the disease and other pest issues. If the right treatment is implemented, the losses value can be lessened to a few thousand ringgit per hectare and will benefit all sectors along the oil palm supply chain. Immediate response and actions need to be taken on the factors discussed such as developing methods through advanced molecular technologies, high-throughput device systems for field use, biochemical tests, generation of resistant genotypes, use of chemicals, cultural practices, and application of “microbial-cocktails”, which provides Ganoderma with a broad antibacterial spectrum through a more affordable, efficient, and environmentally friendly approach [102].

The oil palm industry may learn from its past and present inadequacies in managing sustainability and apply those lessons going forward. Denial and counter-complaints are thought to be less helpful than an open response to criticisms of the sector’s shortcomings. The distinctive qualities of oil palms, such as their high productivity and long lifespan, distinguish them from other oil crops. An overall understanding of all the possible factors based on the plant disease triangle concept should be studied to fully grasp the root cause and effect and possible treatments to be taken for overcoming BSR disease caused by *Ganoderma*. Most research focused on only the factor-and-effect of single main factors from the disease triangle as their main objectives. Thus, this became the main aim of the paper with anticipations to help planters, organizations, and government to overcome the problem, which is by looking at all possible factors that may affect the *Ganoderma* BSR disease progress.

## Figures and Tables

**Figure 1 plants-11-02462-f001:**
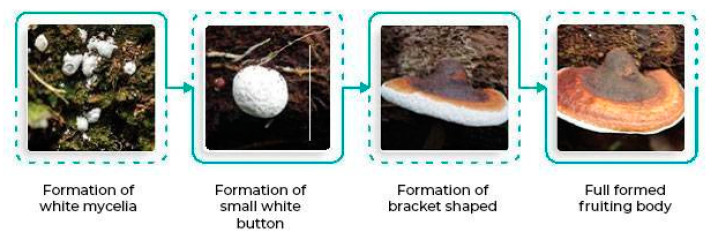
Development of the *Ganoderma* fruiting body.

**Figure 2 plants-11-02462-f002:**
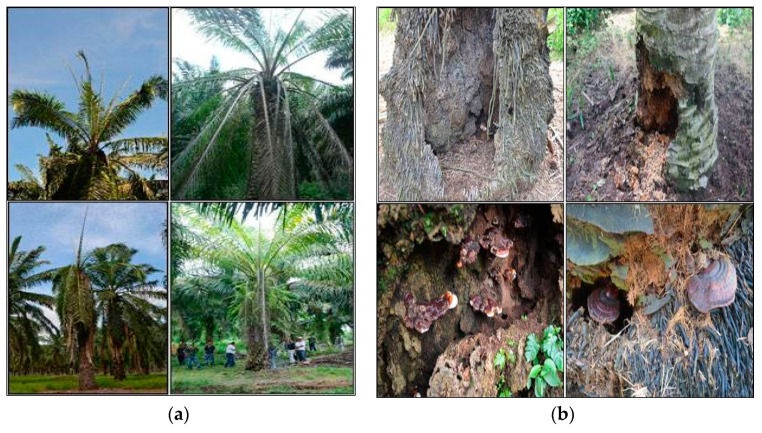
Some of the symptoms of oil palm trees affected by *Ganoderma* BSR disease. (**a**) Two or more unopened spear leaves, yellowing of leaves, and dried fronts; (**b**) stem and bole rotting at the base of the palm, and bracket-shaped form of *Ganoderma* pathogen at the stem and roots of the palm.

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
