# Peer review of "A Review of Factors Affecting Ganoderma Basal Stem Rot Disease Progress in Oil Palm"

_plants, 2022, doi:10.3390/plants11192462_

Round 1

Reviewer 1 Report

Line 74: "elimination", you cannot eliminate any of the factors, just moderate or change.

Line 136: "climate will deteriorate for oil palm"; can you guys elaborate more (i.e. increased temp, higher frequency of drought, etc)

Although biocontrol is mentioned, no examples or discussion is presented.  For long-lived trees like palms, biocontrol should play a key role in extending the useful life of the plants.  How about newer technologies (mRNAi's, chitosan application, etc.).  The authors should be more bold in exploring the future, not just fatalistic about climate change.

Also, the conclusions should shed some light about what has been discovered so far, and what needs to be done.  I do not see that in the conclusions.

Author Response

Attached is the response to the reviewer's comments/ suggestions

Reviewer 2 Report

In the manuscript “A Review of Factors Affecting Ganoderma Basal Stem Rot Disease Progress in Oil Palm” the authors intent to review and clarify the factors responsible of the BSR disease based on the concept of plant disease triangle in order to overcome BSR disease caused by Ganoderma spp. However, the manuscript does not contribute to resume the state of the art about this subject.

On the other hand, besides the high economic importance of the oil palm tree as a valuable crop, the present manuscript that intent to understand the BSR through a holistic manner omitted the net decrease in ecosystem function associated with oil palm tree exploitation and its impact to the loss of rainforest, biodiversity, and soil carbon as well as greenhouse gas emissions that are often referred (eg. ref. 10.3390/microorganisms7010024 and references herein),

This fact is also missed when the authors referred the reported called Agriculture 4.0 Since this report take special attention to Demographics, Scarcity of natural resources, Climate change, and Food waste so the effect of oil palm tree plantations need to be seem in this framework and with a sustainability vision, which is avoid by the authors.

Beside that the manuscript presented defective literature survey that fail to mention the last findings about the pathogen, the oil palm tree and the interaction.

The same happens with the concepts of plant resistance and plant susceptibility, that are not clearly explained.

The generalization made about the reproduction methods of Basidiomycetes is not correct, and must be reformulated or removed!

A figure with a set of pictures should be included to illustrate the disease symptoms along the infection process.

The English should be review.

Author Response

(The authors gave the same response as above.)

Reviewer 3 Report

In general, the MS is wordy and minor revisions are needed.

line 24 change "whereby" to "with"

line 82 add "and" before "thus"

line 201 How can "healthy stumps" be "BSR sources"?

line 212 delete "ones"

line 232, 438 spell the full species name of "b"

line 237 delete "s'"

line 243 delete "s" from "bacterias"

line 269 delete "increased and"

line 278 change "resisting" to "resistant"

line 291 add "ed" to "exist"

line 301 add "was" before "neither"

line 306 delete "in"

line 349 delete "to be looked out for in this factor" and start the sentence with

            "Significant elements"

line 359 add "were" before "made"

line 341 add "and" before "thus" and change "inoculums" to "inocula"

line 437 delete "on"

line 440 delete "It is not unexpected that" and start the sentence with "Finding"

line 441 delete "as"

line 442 delete "to be worked on

Author Response

(The authors gave the same response as above.)

Round 2

Reviewer 1 Report

I still think you should improve your conclusions to summarize major trends in the control and explore what the future research avenues will be.

Point 3 and 4 need to be addressed.  You wrote N/A which I understand it means Not Applicable.  I can tell, it is applicable for a good review of a topic like this.

Reviewer 2 Report

A scientific report about oil palm cultivation should refer the all points of view, so although the authors had inserted the suggested reference (about the environment impact of oil palm plantations) the authors had avoid to include the global concern around oil palm cultivation expansion, namely its role in deforestation; negative environmental impacts including biodiversity declines; greenhouse gas emissions and air pollution, that should be also included.

The authors did not answered to Point 5

Point 5: The concept of plant resistance and susceptibility are not well explained.

Point 5: N/A

The paragraph that starting in line 336 ending at line 346 should include the actual that plants are continuously challenged by several of pathogens that had led to the develop of special mechanisms for pathogen recognition including signalling transduction pathways as a way to deploy the defence responses. On the other side phytopathogens have evolved to manipulate host plants allowing theirs exploitation. The success of the infection depends on how quickly and efficiently the pathogens are recognized and how rapidly the plant response is initiated.

Besides that the authors had  included some references in the text, which I think is in disagreement with the Journal rules, eg. line 48-51; line 60-61; line 356-357; line 360-361; In line 55 the authors could consider the removal of authors of article replacing the names by et al.
